# Root Exudates Mediate the Processes of Soil Organic Carbon Input and Efflux

**DOI:** 10.3390/plants12030630

**Published:** 2023-01-31

**Authors:** Xue Lei, Yuting Shen, Jianing Zhao, Jiajia Huang, Hui Wang, Yang Yu, Chunwang Xiao

**Affiliations:** 1College of Life and Environmental Sciences, Minzu University of China, Beijing 100081, China; 2State Key Laboratory of Vegetation and Environmental Change, Institute of Botany, Chinese Academy of Sciences, Beijing 100093, China; 3University of Chinese Academy of Sciences, Beijing 100049, China

**Keywords:** rhizosphere, root exudates, soil organic carbon, soil microbes

## Abstract

Root exudates, as an important form of material input from plants to the soil, regulate the carbon input and efflux of plant rhizosphere soil and play an important role in maintaining the carbon and nutrient balance of the whole ecosystem. Root exudates are notoriously difficult to collect due to their underlying characteristics (e.g., low concentration and fast turnover rate) and the associated methodological challenges of accurately measuring root exudates in native soils. As a result, up until now, it has been difficult to accurately quantify the soil organic carbon input from root exudates to the soil in most studies. In recent years, the contribution and ecological effects of root exudates to soil organic carbon input and efflux have been paid more and more attention. However, the ecological mechanism of soil organic carbon input and efflux mediated by root exudates are rarely analyzed comprehensively. In this review, the main processes and influencing factors of soil organic carbon input and efflux mediated by root exudates are demonstrated. Soil minerals and soil microbes play key roles in the processes. The carbon allocation from plants to soil is influenced by the relationship between root exudates and root functional traits. Compared with the quantity of root exudates, the response of root exudate quality to environmental changes affects soil carbon function more. In the future, the contribution of root exudates in different plants to soil carbon turnover and their relationship with soil nutrient availability will be accurately quantified, which will be helpful to understand the mechanism of soil organic carbon sequestration.

## 1. Introduction

The soil carbon stock is more than twice the atmosphere carbon stock and plays a critical role in the global carbon (C) cycle [1]. Soil can sequester and store organic carbon, thereby reducing the amount of carbon in the atmosphere, and thus mitigating climate change. However, one concern is that the mineralization of soil organic carbon increases atmospheric CO_2_ concentrations. Soil is a non-homogeneous system, and the temporal scales of different forms of soil carbon turnover considerably vary under the influences of climate change, soil physicochemical properties, land management, and other factors. Due to the limits in monitoring, verification, sampling, and depth methods, it is also difficult to interpret and quantify a small increase or decrease in the soil carbon stock at different spatial and temporal scales and to assess the carbon sequestration potential of soil [2]. The source forms of soil carbon can be divided into directly fixed soil inorganic carbon (such as calcium and magnesium carbonate) and indirectly fixed soil organic carbon (SOC). SOC is a dynamic organic continuum. In order to better understand the underlying processes of SOC formation and mineralization, Lavallee et al. (2020) suggested that SOC should be separated into particulate organic carbon (POC) and mineral-associated organic carbon (MAOC) [3]. POC is mainly derived from litter and root inputs, while MAOC is mainly microbial derivatives [4]. MAOC usually can keep longer [3]. With the deepening of the research on the soil carbon cycle, the mechanism of SOC stabilization has been gradually revealed. Plant carbon inputs are the dominant organic constituent of SOC, and their relative importance outweighs other abiotic factors to SOC stabilization in the topsoil at regional and global scales [5]. Plant inputs enter the soil as either aboveground C or belowground C and include both structural litter inputs (e.g., root and shoot detritus) and dissolved organic carbon (DOC) (rhizodeposition and leaf litter leachate). A suite of recent studies has found that belowground carbon inputs have a greater impact on the efficiency and stabilization of soil carbon inputs than aboveground carbon inputs do [6,7,8]. Belowground carbon inputs retained in the mineral soil are between two and five times more efficient than aboveground carbon inputs at a certain spatial and temporal scale [9]. Plant roots are the key structure connecting the aboveground and belowground parts of plants, and their morphological structure, physiological metabolism, and other characteristics affect the carbon allocation from plants to soil. 

Although soil carbon inputs from living roots account for a small proportion of total plant photosynthate compared to those from litter, they determine the asymmetry of aboveground and belowground SOC distribution and are more effectively retained in MAOC and POC than litter carbon inputs [6]. However, the release of organic compounds into the soil by living roots has been the most uncertain part of the soil carbon flux and cycle [10]. Root carbon inputs can lead to SOC increase and promote SOC stabilization. For example, compared with plant shoot and mycorrhizal pathways, plant roots in temperate steppe contribute to the newest SOC formation under different soil fertility levels [11]. However, root carbon inputs can also reduce SOC stabilization so that preserved C can be used by soil microbes, leading to SOC loss. Meanwhile, plant roots can also reduce the stability of organic carbon through the destruction of aggregates [12]. Therefore, it is very important to identify the mechanism of root carbon inputs on soil carbon stabilization and to evaluate SOC input and efflux with precise quantification. In the root system, fine roots have been the focus of most studies, because they are generally essential for many root functions (e.g., root elongation, nutrient and water acquisition, association with symbionts, and carbon exudates). The roots of grassland plants form a dense network of fine roots in the deeper soil and can alleviate the decomposition of SOC to some extent [13]. The links among fine root classes (including individual root orders), traits, and functions will be a key research area, as morphological and chemical traits of fine roots have been extensively studied, while physiological traits have rarely been considered [14]. 

Rhizodeposition is formed when plant roots release organic and inorganic compounds in the roots to the adjacent soil, which mainly comes from plant detritus, the decomposition and renewal of root cells, and root exudates. Gross rhizodeposition carbon accounts for 5–20% of photosynthates [15]. Root exudation can account for up to 2–11% of the total photosynthetic production [16]. Root exudates are an important part of rhizodeposition, reflecting a physiological property of fine roots, and some dissolved organic materials in root exudates produced by living plants through primary metabolism, secondary metabolism, and abiotic metabolism are the main carriers for plant roots releasing organic carbon into the soil [17]. CO_2_ in the atmosphere that is fixed by plants through photosynthesis reaches the soil via aboveground plant residues, belowground root exudates, and root turnover, and these constitute the main sources of soil carbon. The contribution rate of root exudates and litter to soil organic carbon input in different degraded grassland ecosystems in Inner Mongolia ranged from 8.8 to 14.8% and 16.8 to 17.2%, respectively [18]. While interacting with biotic and environmental factors, root exudates also affect SOC efflux [19]. Most of our previous studies focus on root exudates of the crop plant (bean, wheat) to develop agriculture and the interaction between root exudates and the rhizosphere environment to reveal the relationships between root exudates and soil carbon and nitrogen. With the increasing development of basic research content and analytical methods, soil chemistry and ecology have gradually become the cross-content of root exudates research, and the research topic has gradually become ecological problem mechanisms from the application [20]. The processes of the soil carbon cycle mediated by root exudates and its influencing mechanism are key and difficult points in the construction of the global carbon cycle model [21]. In many studies, the physiological characteristics and ecological effects of root exudates have been summarized [22,23], and their response to stress in the environment and feedback role [24,25] and the influence of root exudates on SOC input and carbon efflux have also been mentioned [12,26], but few studies have integrated SOC input and efflux to explore root exudates. In order to quantify the contribution of abiotic factors (e.g., physical and chemical factors) and biologic factors (e.g., microbial community factors) to soil carbon mineralization and stability mediated by root exudates, and to clarify the potential mechanism of root exudates in soil carbon sequestration, fully understanding the processes of SOC input and efflux mediated by root exudates and their effect mechanism is important. The processes of SOC input and efflux mediated by root exudates are illustrated and the mechanism of root exudates on SOC sequestration is revealed in this paper (Figure 1). The current research hot spot and the aspects needed to be further studied are reviewed, which help to accurately quantify the contribution of root exudates to SOC input and efflux, predict the variation of soil carbon stock, evaluate the ability of soil carbon sequestration, and provide the scientific basis for improving potentiality of soil carbon sequestration.

(i): The process of SOC input in which root exudates are involved: root exudates form dissolved organic carbon (DOC), and then form MAOC through the mineral binding pathway; root exudates are decomposed and consumed by microbial activity to form MAOC. (ii): The process of SOC efflux in which root exudates are involved: root exudates are decomposed and utilized by microbes and promote the decomposition of soil stable organic carbon; root exudates release the protected C from the soil through the physical chemistry of coordination complexation and dissolution and can be decomposed and utilized by microbes.

## 2. Root Exudates and Soil Organic Carbon Input

The interactions of root exudates with soil microbes and minerals are beneficial to SOC formation. Active compositions in root exudates effectively form MAOC by combining with organic minerals in the surrounding mineral soil [27]. Unstable DOC (such as monosaccharides) compounds in root exudates provide soil microbes with important precursors involved in SOC formation [28]. Firstly, there is spatial heterogeneity in SOC formed by root exudates in the rhizosphere. The effects of dissolved matter in root exudates on SOC storage and structural development are significantly different between surface and bottom soils, and the potential for SOC accumulation in deeper soil is higher than in surface soil due to the addition of root exudates. In forests, a large number of dissolved matter in root exudates can induce large aggregation of organisms and storage of carbon in deeper soil [29]. Secondly, the composition complexity of root exudates affects the formation of different soil carbon components. Simple compositions in root exudates may promote the formation of MAOC, while complex compositions may promote the formation of POC [30]. So, the same compositions of root exudates have different effects on SOC stability in different spaces, and the different quantities and compositions of root exudates have different effects on soil carbon turnover. 

The quantity and quality of root exudates vary significantly among plant species [31]. Phylogeny is a common controlling factor of root functional traits and root exudates [32]. In recent years, root exudates have been gradually taken into account in the construction of multi-dimensional traits and economic spaces to study the functional mechanisms of roots. Root functional traits can reflect the characteristics of root morphogenesis, chemical metabolism, and symbiosis. For example, specific root length is commonly known as specific root area and is a compound trait determined by root diameter (RD) and root tissue density (RTD), which reflects the cost of root construction; root nitrogen concentration (RNC) is used to reflect the overall metabolic activity of roots; and mycorrhizal colonization represents the symbiotic character of plants. All these traits reflect the strategies and efficiency of plant uptake of soil nutrients [33]. The root exudation rate has been shown to have different correlations with root functional traits in different species, and it can be used as a functional trait and a key indicator of the plant resource utilization strategy (utilization and conservation) [34]. For instance, root exudates from the fine roots of woody plants are positively correlated with root N concentration [35]. Some studies also report that the specific root exudation rate increases with the increase in root diameter and decreases with the increase in RTD, but it has no relationship with RNC [36]. Evidence from studies on root exudates suggests that the root exudation rate is a survival strategy for plants to adapt to a low-nutrient environment, and an increase in the root exudation rate leads to a high cost of root construction and slow root growth [35]. There is a negative correlation between mycorrhizal colonization and the root exudation rate, indicating that there is a trade-off between mycorrhizal symbiosis and root exudates in belowground carbon allocation [37]. Based on the analysis and research on the characteristics of root exudates, the quality of root exudates (metabolic characteristics) is considered to play a more important ecological role in root functional traits. RD, RTD, and RNC change obviously in the metabolic composition of root exudates [38]. The composition of primary metabolites in root exudates varies with the plant growth strategy, and the root of more competitive species (faster-growing species) secretes more primary metabolites (e.g., sugars, organic acids), while the root of conserved species (slower-growing species) secretes more amino acids [34]. Therefore, root functional traits determine the carbon allocation of root exudates and nutrient utilization strategies of plants in different soil stocks, and the relationship between root exudates and root traits mainly depends on plant characteristics. 

## 3. Root Exudates and Soil Organic Carbon Efflux

It is generally believed that the increase of SOC input can effectively promote soil carbon sequestration, but more and more evidence shows that SOC input will also change the turnover of the original SOC, thus resulting in SOC loss. Root exudates, as an input form of active C substrate, can stimulate the decomposition of more stable SOC [39], which is a process known as the “priming effect”. One mechanism is the “Microbial metabolism theory”, in which root exudates, as the energy matrix of soil microbes, stimulate the decomposition of SOC by affecting the structure, composition, and activity of microbial communities. Another new mechanism is that root exudates (oxalic acid) disrupt or reduce the stability of the MAOC complex by physical chemistry, thus allowing the release of protected C from the soil for microbial decomposition and utilization [19]. There is no consensus on the mechanism of the rhizosphere priming effect. Although the priming effect has been proven to be prevalent in nature, most relevant studies are currently conducted in greenhouses. It is indeed necessary to strengthen field experiments as well as long-term observation and data accumulation [39]. Frequent carbon inputs can induce a stronger priming effect than occasional carbon inputs do, so increasing carbon input frequency may weaken the carbon sink of grasslands [40]. This means that the priming effect is likely to offset the contribution of root exudates to SOC input, and the involvement of root exudates in the formation of SOC does not improve the carbon sequestration potential of soil to some extent as a whole, but only provides a short-term prediction.

The quantity and composition of root exudates are different and make the intensity of the priming effect different. Higher rhizodeposition can accelerate soil carbon turnover [41]. The simpler the structure of the root exudates C matrix, the faster the energy conversion, the more complex the structure of the root exudates C matrix, and the less the priming effect [42]. Plant functional traits are key factors for the priming effect to affect SOC decomposition. An increase in the root exudation rate will increase the influence of the priming effect on organic matter decomposition [43]. Due to differences in the strategies and efficiency of using C by different soil microbes, the distribution and turnover of root exudates in different soil microbial groups are different, and different bacterial functional groups in the soil have different effects on the priming effect. According to the relative distribution and turnover rate of root carbon deposition in the individual microbiome in the soil, the mean turnover rates of rhizodeposited C in microbial groups ranged from 0.04 to 0.13 day^−1^. Gram-negative bacteria and fungi were the dominant microbiomes that converted root C deposition into SOC [15].

The stability and activity of SOC also affect the intensity of the priming effect. SOC stability (characterized by chemical resistance and physical chemistry protection) can better account for differences in the priming effect than the plant, soil, and microbial characteristics [44]. The priming effect of soil carbon increases with an increase in soil refractory carbon components but decreases with an increase in soil aggregate and mineral protection [45]. The relatively more stable SOC has a higher priming effect than the unstable SOC does. The rapid turnover of root exudates mainly depends on material decomposition and microbial utilization. Root exudate decomposition is an important part of the carbon cycle in terrestrial ecosystems. Most rhizodepositions, especially active C in root exudates easily used by soil microbes, have an important impact on soil C dynamics [46]. However, root exudates such as oxalate may also release organic compounds directly from organic mineral aggregates, so that soil microbes access these compounds easily, resulting in increasing net soil carbon loss [47]. The part of the carbon released by microbial decomposition in the priming effect is not released into the atmosphere, and this part of carbon can compensate for the carbon loss caused by the priming effect, but where the carbon goes in the soil is still uncertain [48]. Some studies have quantified the decomposition rate of SOC in the rhizosphere of grasslands and their responses to drought and nitrogen addition using ^13^C-CO_2_ pulse labeling. The results show that the decomposition amplitude of accumulated rhizodeposition C was similar to that of accumulated SOC. Decomposition of rhizodeposited C accounted for 7–31% of the total below-ground respiration in grassland ecosystems. It may be a larger component of CO_2_ released from the soil in grasslands [49]. Therefore, it is necessary to further improve the understanding of the influence of abiotic factors on the intensity and direction of the priming effect.

Although the interaction between root exudates and biotic environmental factors can promote the decomposition rate of SOC, this process improves soil nutrient availability to a certain extent. The priming effect indicates that root exudates mobilize nutrients from mineral assemblages or soil parent material [19], or accelerate microbial decomposition [50] and promote the nutrient status of the soil. A significant proportion of photosynthetic C is released into the soil as rhizodeposition (including exudates, exfoliated cells, and mucus), stimulating microbial activity in the decomposition of SOC and nutrient release [51,52]. The rate and composition of root exudates can increase nutrient mineralization by soil microbes and affect plant recovery under drought stress [38]. According to the “stoichiometric decomposition” theory, soil N availability in turn influences the ratio of C to N in root exudates to match the needs of soil microbes and then influences the decomposition rate of SOC stimulated by soil microbes [26]. Root exudates affect SOC efflux through an interaction with the soil biologic environment, and they also interact with other pathways of SOC input and efflux, which affect SOC efflux. The non-additive effect of root exudates and litter inputs may accelerate SOC formation and SOC consumption [6]. It has also been further shown that rhizodeposition (at least root exudates) and root respiration are likely to come from the same unstructured C pool, and they have similar turn-around times [53,54]. In the study of predicting SOC flux, when climatic conditions change, unstable carbon inputs in root exudates can improve the temperature sensitivity of soil microbial respiration with low substrate utilization in soil [55]. Shen et al. (2020) found that in the process of grassland degradation, root biomass and plant litter decreased, but the root exudation rate increased and the relative contribution of root exudates to SOC input increased, which would further affect the formation and stability of SOC in grasslands [18].

## 4. Mechanism of Root Exudates Affecting Soil Organic Carbon Sequestration 

Soil organic carbon stability mainly depends on the form and persistence of SOC. SOC formation efficiency (SOC_FE_), formerly known as the humification coefficient, is defined as the proportion of carbon inputs retained in SOC and depends on the formation and mineralization rates of the POC and MAOC components [56]. The response of SOC to global change depends on the changes in POC and MAOC [57]. However, the carbon in MAOC and POC have different sensitivity to climate change in different ecosystems [58]. Root exudates form MAOC through the formation of DOC and then through the mineral binding pathway [59]. Some studies have shown that roots and root exudates are efficient carbon sources for POC and MAOC formation, respectively [60]. Root exudates of different plants usually contain compounds such as hydrocarbons, acids, and esters, and the simple carbohydrates in root exudates are easily consumed by microbial activities to produce microbial-derived compounds and to recycle into DOC, which is the main and more efficient pathway for MAOC formation [61]. Studies have shown that root exudates have a positive correlation with SOC accumulation and have a significant impact on grassland biodiversity. Grasslands with high species richness also have a higher amount of SOC accumulation [62]. Moreover, in forest and grassland ecosystems, root exudates, as a source of SOC, can stabilize SOC through various mechanisms and lead to long-term SOC sequestration [63]. However, it is not clear how the contribution of root exudates to SOC accumulation (POC and MAOC) varies with grassland type, soil properties, and climatic conditions [64]. There is little research on the different mechanisms of root exudates impacting soil organic carbon between the two ecosystems. Both plants and soil microbes may be the sources of common compounds in MAOC, and the direct quantitative contribution of root exudates to MAOC is still unclear [65]. To some extent, the increases in exudation also stimulate soil organic matter decomposition, and such changes may prevent soil C accumulation in forest ecosystems [66]. Therefore, root exudates are an important form of SOC input, and in order to understand the effects of root exudates on SOC sequestration, it is necessary to analyze the relationship between root exudates and SOC formation, persistence, and mineralization and the influencing mechanism. The contribution of plant carbon inputs and microbial residues to the soil carbon stock needs to be clarified to analyze the effects of root exudates on the stability of the soil carbon stock.

SOC stability is controlled by biologic factors, such as the efficiency of chemically distinct organic compounds that soil microbes use and produce, and by abiotic factors, such as the adsorption between soil minerals and the organic matter from plants and soil microbes [67]. Soil microbial community structure and diversity have different correlations with SOC in grassland ecosystems with different degradation and management patterns [68,69]. Root exudates, participating in soil carbon turnover through biotic and abiotic pathways, have different effects on SOC stabilization. Specifically, carbon components in root exudates can be combined with microbial metabolites, dead detritus, or soil minerals, and remain in the soil through the physical protection of aggregates or mineral–chemical binding forms. For example, the iron oxide concentration in paddy soils can physicochemically stabilize carbon in corn stover by co-precipitation with rhizodeposition to form the Fe–OM complex [70]. Through the microbial formation pathway, belowground carbon inputs are more effective in forming MAOC than aboveground carbon inputs, in part because rhizosphere microbial communities are more efficient at forming stable soil carbon than bulk soil communities [61]. Compared with the microbial materials produced by root and bud litter, the microbial available materials produced by rhizodeposition are more efficiently converted into MAOC [6]. This is because when organic matter is combined with minerals, the contact between soil microbes and enzymes can be effectively reduced, which then reduces the probability of organic matter being degraded [71]. Since different components in root exudates have different energy content, C oxidation status, and structural complexity, they have different effects on the carbon use efficiency (CUE) of soil microbes [72]. Microbial functional groups with different physiology and abundance have different contributions to the transformation of rhizodeposited carbon into SOC. Compared with bacteria, fungi can produce more microbial residues through rhizodeposited carbon [15]. 

The responses of root exudates to the external environment also affect their effects on SOC input and efflux. The quantity and composition of root exudates have different responses to different environmental factors. Elevated CO_2_ concentration increases the efflux rates of soluble sugars and carboxylate (including citrate) but has no significant effect on amino acids and malate [73]. Under night-time warming, the contents of carbohydrates, amino acids, and phenolic acids in the root exudates of spruce increase, while the contents of lipids and ethers decrease significantly [74]. Under drought conditions, the contents of organic acids, amino acids, and sugars in root exudates increase relatively [75,76]. During the period from drought to drought recovery, the metabolite composition in root exudates is mainly composed of some amino acids, and most sugars and secondary compounds decrease [16]. This change in the composition of the root exudates has an important impact on soil respiratory function [77]. Application of N inhibits the C inputs of root exudates per unit root surface area and significantly reduces the effect of warming on root carbon inputs [78]. Under the influence of global change, the increase in the root exudation rate and the proportion of monosaccharides in root exudates may decrease the soil carbon storage capacity [79]. The quality of root exudates is especially sensitive to soil environmental change compared with the quantity of root exudates, and the relative content of different components in root exudates play a more important role in responses to environmental factors and their interaction, and the responses generally improve the plant resistance and adaptability and directly or indirectly affect the turnover rate of SOC and the processes of SOC input and efflux. 

Root exudates have complex compositions, low content, a fast turnover rate, and a small distribution area in the rhizosphere. In addition, because of the decomposition and utilization of soil microbes and the interaction with soil environmental factors, the influencing factors of root exudates are complex and variable. The quantity, composition, and production mechanisms are not uniform or static properties of root exudates, and they are affected by internal and external factors such as plant species, growth stages, root traits, soil microbes, environmental conditions, nutrient conditions, and soil types [80]. Because of different collection methods and plant species, the samples collected from natural soils may be marred by root damage and the high chemical complexity, which confounds the identification of the origin of derived metabolite sources. Moreover, there are some differences between the experimental environment and the soil environment, which will affect the monitoring of some rhizosphere ecological processes driven by root exudates, resulting in the lack of corresponding comparative analysis [81]. Therefore, it is difficult to study the methods of collecting root exudates. In recent years, an in situ collection device can be more effective without disturbing the natural root growth and damaging the rhizosphere process [82]. However, this method can only monitor and analyze some components of root exudates in real time and cannot identify the information on the whole components of root metabolites, the release mechanism, and their dynamic changes [83]. Therefore, the quantity and component categories of root exudates vary under different conditions and extraction methods, which increases the difficulty of studying the ecological role of root exudates in mediating SOC input and efflux.

## 5. Conclusions and Prospects

Overall, SOC stability depends on the relation between SOC input and efflux, and the increase in SOC input does not represent the increase in soil carbon sequestration. As a part of rhizodeposition, root exudates affect SOC stability through an abiotic–biologic coupling pathway, and the quantity and quality of root exudates are the major determinants of SOC stability by the physicochemical and biological pathways. However, most studies have not separated root exudates from rhizodeposition. Root exudates are an important part of soil carbon inputs, and they can be directly combined with soil organic minerals to form SOC or indirectly participate in the process of microbial formation and then be converted into SOC. Plant species determine the characteristics of root exudates, and environmental factors affect the changes in root exudates, which further affects the contribution of root exudates to SOC formation. Root exudates affect SOC efflux mainly through the priming effect, and the species of root exudates and soil microbes play an important role in the direction and intensity of the priming effect. There are few studies on the relationship between the decomposition of root exudates and the decomposition of SOC. The carbon inputs from root exudates to soil are not equal to their storage in the soil, and the fact that root exudates promote SOC efflux does not mean that they reduce soil carbon sequestration capacity. Different types of root exudates may have different residence times and can be converted into different forms of SOC through abiotic and biologic pathways and then affect the turnover and storage of SOC. Meanwhile, the processes of SOC turnover mediated by root exudates are related to properties of SOC components, soil nutrient availability, and other processes of SOC efflux. Most of the studies on the processes of soil carbon turnover mediated by root exudates focus on grassland and forest ecosystems. While many studies in other ecosystems (farmland, wetland, etc.) pay more attention to the application of root exudates’ characteristics for production practices, the research on root exudates’ influence mechanisms on soil organic carbon turnover is lacking. Therefore, in view of the research progress and deficiency of the processes of SOC input and efflux mediated by root exudates, some prospects for future research on this content are put forward:


(1)The relationship between root exudates and root functional traits in different species


Root exudates are important components of SOC input from plant roots to the soil, and root functional traits reflect plant resource utilization strategies. Root exudates and root functional traits mainly depend on plant species. The root exudation rate, reflecting the adaptive strategies of plants, is gradually incorporated into the study of the multi-dimensional economic spectrum of roots. Different root exudates promote the formation of different soil carbon components and then affect the contribution of root exudates to SOC input. Therefore, the study of the relationship between the quantity and composition of root exudates and root functional traits in different species can help to better predict the relationship between root exudates and SOC input and distribution and to quantify the contribution of root exudates to SOC accumulation under different conditions.


(2)The interaction mechanism between root exudates and soil abiotic factors


Root exudates can cause SOC decomposition through the priming effect. The types and changes in root exudates and soil microbes are very important to understand the intensity and direction of the priming effect. Meanwhile, root exudates can also stimulate SOC efflux through the abiotic pathway, which is related to the mineral form of SOC. It is difficult to quantify the contribution of root exudates to the formation of different carbon components in soil because of their rapid turnover and complex components. Most studies on the relationship between root exudates and SOC stability focus on the biological mechanism, and the abiotic processes and influencing mechanisms of root exudates are often neglected. In order to reveal the influence mechanisms of root exudates on soil carbon components and turnover at different species and community levels, it is necessary to study the spatial distribution and regulation mechanisms of different soil carbon components in the future.


(3)The effects of root exudates on soil carbon function


The carbon inputs from root exudates to soil reflect plant utilization strategies of soil nutrients. Root exudates can improve soil nutrient availability and plant growth, and then promote SOC efflux, so as to improve soil carbon sequestration capacity. In the future, it is necessary to study the processes of SOC input and efflux mediated by root exudates in combination with the species’ characteristics of root exudates and their responses to environmental changes. The contribution of root exudates to different forms of SOC input and the fate of root exudates in soil should be considered. Studies on the relationship between root exudates and soil organic carbon should be analyzed in combination with the application function of root exudates in other ecosystems. Meanwhile, in order to fully understand the long-term mechanism of root exudates on soil carbon function in different ecosystems, the role of root exudates on more ecologically relevant timescales and spatial scales is needed to be quantified. The process of SOC efflux mediated by root exudates should be combined with soil nutrient availability and other processes of SOC efflux.

## Figures and Tables

**Figure 1 plants-12-00630-f001:**
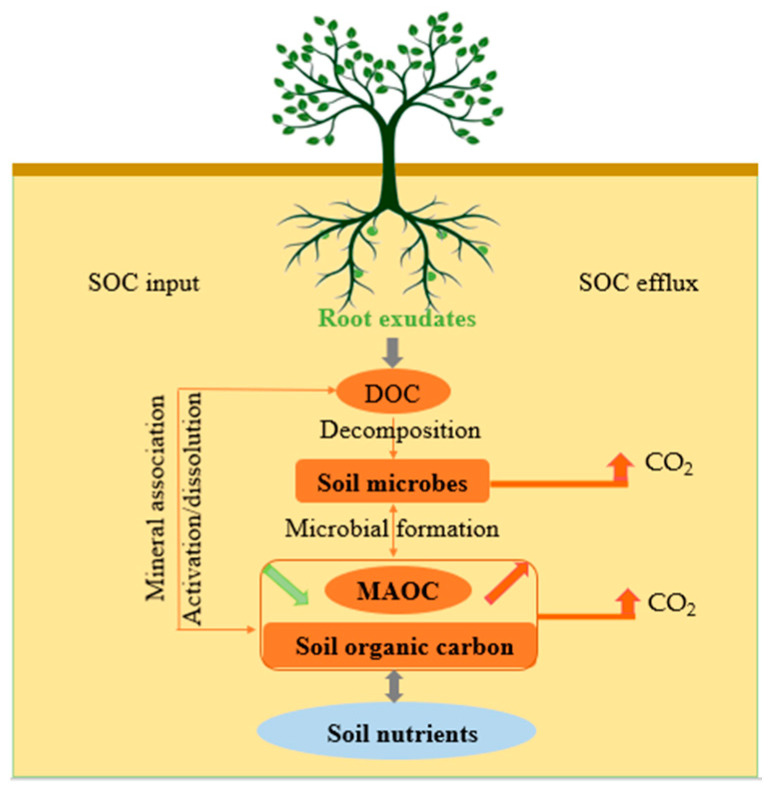
Main processes of soil organic carbon input and efflux mediated by root exudates.

## Data Availability

Not applicable.

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
