# Peer review of "Root Exudates Mediate the Processes of Soil Organic Carbon Input and Efflux"

_plants, 2023, doi:10.3390/plants12030630_

Round 1

Reviewer 1 Report

The problem of identification and quantitative measurement of root exudates (RE) is essential for the theory of the soil organic matter formation and explaining of soil carbon sequestration mechanisms. Therefore, this review article is the next stage for a scientific understanding of the nature and composition of RE and rhizodeposition (RD) as well as their fate in the soil. Of course, there are many more research directions in the controversial and debatable problem of RE than those noted by the authors in this review. Nevertheless, this work is relevant and useful.

In my opinion, the following main questions were not considered in the review:

What are the approximate amounts of carbon input into the soil with both RE and RD?

What is the turnover time of carbon from root exudates in the soil?

Who are the main consumers of RE? Is it bacteria or fungi?

There are also several remarks within the manuscript.

L. 31-32. “Soil can absorb and fix carbon dioxide (CO2) in the atmosphere through plant photosynthesis,”

The soil weakly adsorbs CO2 and it does not fix CO2. CO2 fixation in soil is carried out by autotrophic bacteria and algae. Plants enrich the soil with fixed carbon through plant residues and root exudates.

L. 34. “Soil is non-transparent”.

Soil is non-homogeneous system.

L. 41-42. “SOC is a dynamic organic complex”

SOC is a dynamic organic continuum

L. 47-49. “Plant inputs, the dominant organic constituent of SOC, have a greater impact on SOC stabilization than other abiotic factors at regional and global scales do [5].”

Stabilization of organic matter is controlled by the quality of plant residues and physical and chemical interactions of organic compounds with the mineral matrix.

L. 59-60.  “proportion of total plant photosynthate compared to those from litter”

Could you point out some of these proportions?

L. 67-69. “But root carbon inputs can also reduce SOC stabilization so that preserved C can be used by soil microbes, leading to SOC loss. This is mainly because plant roots can promote the stability of organic carbon by affecting the formation of soil aggregates”.

It seems that these two sentences contradict each other. Root exudates contain polysaccharides and ones contribute to the soil aggregates formation and the physical stabilization of organic carbon.

L. 116-118. The diagram is poorly drawn. This figure creates a false impression of the replenishment of root exudates from the MAOС pool. In addition, the main efflux pathway of RE carbon is the formation of CO2 due the heterotrophic respiration, but it is not shown in the figure.

L. 175-176. “but more and more evidences show that SOC input will also change the turnover of original SOC, resulting in SOC loss”

Yes, the priming effect is real, but quantitatively insignificant in comparison with the carbon stocks in the soil. See line 260

L. 176-177. “Root exudates, as an input form of active C substrate, can stimulate the decomposition of more stable SOC”

This conclusion is just a theoretical picture from pure experiments, but in reality it is not realized due to effect of external factors. In one of the experiments, the amount of CO2 released from soil with plant residues did not differ significantly from incubation with vermiculite (DOI: 10.1134/S1064229319100119).

L 194-195. “The increase in root exudation rate will increase the influence of the priming effect on organic matter decomposition”

However, the more root exudation, the more soil microorganisms will be provided with carbon, the less they need for soil carbon.

L. 202-203. “SOC stability (characterized by chemical resistance and physical chemistry protection)”

Please keep in mind that in soil there are also biological stabilization mechanisms for organic matter.

L. 323-324. “Root exudates have complex compositions, low content, fast turnover rate, and a 323 small distribution area in the rhizosphere”

Actually, bulk soil is not enriched with root secretions. Therefore, the priming effect of root exudates can be characteristic only for a limited rhizosphere soil area.

With best wishes in New Year from the Reviewer!

Reviewer 2 Report

This manuscript is a review of the state of knowledge regarding the relationship between root exudates and soil organic carbon fluxes. Unfortunately, this review lacks direction. The authors fail to synthesize the existing research into meaningful information. Most of the manuscript is a collection of individual statements summarizing a specific reference. For example, line 221: "Rhizodeposition C decomposition may be a larger component of CO2 released from soil in grassland." Why? What is the mechanism? How do other systems differ? Many of these statements completely lack any type of discussion highlighting their importance or frequency. Another is line 232, which mentions drought stress. This is the only mention of drought in this entire section. Why? Drought is discussed later in lines 308+. Effects of plant stress on the relationship could be its own section. Line 240 talks about turnaround times but does not provide these times. Many of the examples provided are from grasslands. How do these observations differ in different systems? Occasionally a different system is mentioned (forest, rice paddy, etc.) but that does not indicate if these observations are similar across environments, how they differ, or even if anyone know what they are yet. There are many other examples of important statements being made with no other related information being provided or a synopsis of missing knowledge that should be addressed. To be a useful review, it needs to be broken into the key factors influencing root exudates (plant, soil, climate, etc.), the relationships between root exudation and SOC, and what the state of knowledge is regarding factors influencing those relationships, followed by future directions. In its current form the manuscript provides little new information. 

Reviewer 3 Report

Dear Authors,

The publication  contains important and valuable information for gardening, agriculture and forestry.  I believe that the publication should be accepted for publication. The only remark is the large number of abbreviations. They should be explained in one place, at the beginning or at the end of the article, so that the reader can easily find and decipher them. I believe that the publication should be accepted for publication. The only remark is the large number of abbreviations. They should be explained in one place, at the beginning or at the end of the article, so that the reader can easily find and understand them.  Such a large number of abbreviations sometimes makes the article unreadable.
The literature inventory has been conscientiously prepared. It includes the most recent publications on the topic covered.

Round 2

Reviewer 2 Report

The manuscript is much improved, although the authors still fail to expand the level of knowledge beyond what has been published. Simple statements such as "research into other systems are lacking" or "relationships beyond those noted here have not been conducted yet" need to be included to inform readers that much work is yet to be done and you are basing your statements off of incomplete information. 

Title should be "mediate" not "mediated". The processes are still occurring and will continue to. 

Verb tenses and other grammatical issues still remain throughout the manuscript. 

Line 366: only grasslands are mentioned. In line 371 forest ecosystems are mentioned. What about the other biomes. When just a single or few biomes are mentioned it seems like this process is only important in them. You need to include a statement that these relationships are unknown in other systems. 
